# Social support receipt as a predictor of mortality: A cohort study in rural South Africa

David Kapaon[1]*, Carlos Riumallo-Herl[2,3], Elyse Jennings[1], Shafika Abrahams-Gessel[4], Keletso Makofane[5], Chodziwadziwa Whiteson Kabudula[3], Guy Harling[1,3,6,7,8]

1 Harvard Center for Population and Development Studies, Harvard T.H. Chan School of Public Health, Boston, Massachusetts, United States of America, 2 Erasmus School of Economics, Erasmus University Rotterdam, Rotterdam, The Netherlands, 3 MRC/Wits Rural Public Health and Health Transitions Research Unit (Agincourt), School of Public Health, Faculty of Health Sciences, University of the Witwatersrand, Johannesburg, South Africa, 4 Harvard Center for Health Decision Science, Harvard T.H. Chan School of Public Health, Boston, Massachusetts, United States of America, 5 FXB Center for Health and Human Rights, Harvard University, Boston, Massachusetts, United States of America, 6 Institute for Global Health, University College London, London, United Kingdom, 7 Africa Health Research Institute, KwaZulu-Natal, Durban, South Africa, 8 School of Nursing & Public Health, College of Health Sciences, University of KwaZulu-Natal, Durban, South Africa

* kapaondavid@gmail.com

**Data Availability Statement:** HAALSI Public-use datasets can be accessed from the following three institutions: 1. The Institute of Quantitative Social Science's Dataverse at Harvard University (IQSS:

## Abstract

The mechanisms connecting various types of social support to mortality have been well-studied in high-income countries. However, less is known about how these relationships function in different socioeconomic contexts. We examined how four domains of social support—emotional, physical, financial, and informational—impact mortality within a sample of older adults living in a rural and resource-constrained setting. Using baseline survey and longitudinal mortality data from HAALSI, we assessed how social support affects mortality in a cohort of 5059 individuals aged 40 years or older in rural Mpumalanga, South Africa. Social support was captured as the self-reported frequency with which close social contacts offered emotional, physical, financial, and informational support to respondents, standardized across the sample to increase interpretability. We used Cox proportional hazard models to evaluate how each support type affected mortality controlling for potential confounders, and assessed potential effect-modification by age and sex. Each of the four support domains had small positive associations with mortality, ranging from a hazard ratio per standard deviation of support of 1.04 [95% CI: 0.95,1.13] for financial support to 1.09 [95% CI: 0.99,1.18] for informational support. Associations were often stronger for females and younger individuals. We find baseline social support to be positively associated with mortality in rural South Africa. Possible explanations include that insufficient social support is not a strong driver of mortality risk in this setting, or that social support does not reach some necessary threshold to buffer against mortality. Additionally, it is possible that the social support measure did not capture more relevant aspects of support, or that our social support measures captured prior morbidity that attracted support before the study began. We highlight approaches to evaluate some of these hypotheses in future research.

https://www.iq.harvard.edu/), 2. The University of Michigan's Inter-University Consortium for Political and Social Research (ICPSR: https://www.icpsr. umich.edu/web/pages/), and 3. The INDEPTH Network data repository (http://www.indepth-network.org/).

**Funding:** HAALSI is supported by the National Institute of Aging of the National Institutes of Health (P01AG041710 to LFB). HAALSI is nested within the Agincourt Health and socio-Demographic Surveillance System with funding from the South African Department of Science and Innovation through the South African Population Research Infrastructure Network (SAPRIN), the University of the Witwatersrand, the South African Medical Research Council, and previously the Wellcome Trust [Grant numbers 058893/Z/99/A; 069683/Z/02/Z; 085477/Z/08/Z; 085477/B/08/Z]. GH was supported by a fellowship from the Wellcome Trust and the Royal Society [210479/Z/ 18/Z]. CRH received funding from the European Union's Horizon 2020 research and innovation programme under the Marie Sklodowska-Curie grant agreement No 840591. For the purpose of open access, the authors have applied a CC BY public copyright license to any Author Accepted Manuscript version arising from this submission. The funders had no role in study design, data collection and analysis, decision to publish, or preparation of the manuscript.

**Competing interests:** The authors have declared that no competing interests exist.

# 1. Introduction

As individuals worldwide live longer, all countries are experiencing shifting demographics skewed towards increasingly aging populations [1, 2]. Furthermore, the world's population of individuals 60 years of age or older is estimated to reach 2 billion by 2050 with approximately 80% living in low- and middle-income countries [1, 2]. Although a robust set of literature exists on the relationships between individual health behaviours and successful aging, social support also has implications for health and well-being later in the life course as well [3]. However, most of these past studies examine populations from high-income countries, and there is very little research on the associations between social support and mortality in low-income settings. Among a population-representative cohort of adults 40 years of age or older living in a very low-income, rural setting in Mpumalanga Province, South Africa, we analysed associations between four domains of social support and mortality over a five-year period.

## 1.1 Background

**1.1.1 Social support.** Social support can be defined as the "perception or experience that one is cared for, esteemed, and part of a mutually supportive social network, [that] has beneficial effects on mental and physical health" [4]. Typically, it is categorized into one of four domains: Emotional, Instrumental/Physical, Informational, and Appraisal/Companionship [4–6]. Emotional support normally refers to feelings of gratitude, love, and trust, which are exchanged between individuals. Instrumental or physical support usually denotes 'tangible aid', such as material or monetary services, while Informational support includes 'intangible aid', like guidance or suggestions.

These support domains can act singly or in concert to provide mental and physical health benefits to individuals. Furthermore, the mental and physical health effects stemming from each support type can operate socially through personal relationships and resources which combine to help maintain healthy behaviours, or biologically through decreases in cortisol and increases in oxytocin [7, 8]. From both perspectives, two broad mechanisms are theorized to connect social support to mortality: direct effects and buffering [5, 9, 10]. The direct effects' hypothesis highlights the continuous protective benefits that social relationships provide during both stressful, and non-stressful times [4]. On the other hand, the buffering hypothesis argues that social support primarily mediates negative health effects during times of more intense mental or physical anxiety, and offers little, if any, value during non-stressful times [4].

Furthermore, there is ample evidence demonstrating how the quantity and quality of social support an individual has access to can be linked to both positive and negative health outcomes, including longevity and survival [11–13]. After early work documented the importance of social ties in predicting morbidity and mortality [14–17], social support has been shown to increase the uptake of, and adherence to, positive health behaviours including exercise and smoking cessation, as well as reduce mortality from chronic conditions including diabetes, hypertension, and myocardial infarctions [18]. However, while the downstream effects of social support are typically thought of as universally positive, some research has studied how negative interactions could theoretically be more predictive of mortality than positive ones [19–21]. Regardless, almost all of these findings arise from higher-income countries, and it is not clear whether the mechanisms through which social support is believed to affect health will act in the same way, or to the same degree, in low- or middle-income settings, like South Africa [11].

**1.1.2 Social support in South Africa.** The population in question for this analysis lives in a cluster of villages called Agincourt, located within the Bushbuckridge sub-district of rural,

Mpumalanga Province, near the border of Mozambique. Agincourt, as well as South Africa in general, provide an apt context to examine these associations for a number of reasons.

First, many residents are still dealing with the lingering effects of Apartheid-era policies which not only generated distinct morbidity and mortality patterns, but also created stark income disparities given limited investments in public social welfare programs, especially in rural areas like Agincourt [22–26]. While South Africa's inequality did not begin with Apartheid, virtually all of the policies enacted during its existence restricted access to jobs, movement, capital, land, political rights, and resources for Black South Africans [27]. In turn, this strict system of 'separateness' prevented Black South Africans from accumulating wealth or political power, further deepening the already unequal income distribution [28]. Importantly, past research has theorized how high levels of income inequality can impact social support via reduced levels of 'social cohesion' or 'social capital', which are often thought of as the various social organizations, norms, and values that help create a mutual sense of community–(i.e. trust, reciprocity, and civic participation) [29–35]. These norms often pattern how social networks operate, and consequently, how social support is given and received. Similarly, others have postulated that wide disparities between the rich and poor also correlate with general under-investments in the types of infrastructure that help create 'human capital', like schools or hospitals, which also impact health and health outcomes [30, 32, 35, 36]. In particular, this overall lack of formal support infrastructure can place greater demands on leveraging interpersonal relationships to supplement gaps in the social safety net, especially in low-income settings where material or financial resources are scarce [37]. Therefore, we hypothesize that, of the four domains of social support, receipt of financial support will be the most predictive of survival given that higher quantities of financial support will likely have important consequences in a resource-constrained setting like Agincourt.

Second, South Africa's high levels of income inequality have also reshaped the social fabric of the country too, creating unique migration patterns that have important implications for gender roles, the structure of social networks for men and women, as well as the relationship between social support and mortality [38–40]. For example, many Apartheid-era policies impacted movement and work, which in-turn necessitated migration for living arrangements or employment [41–46]. In particular, rural couples often had to live separately to abide by laws that only allowed Black South Africans to live in cities if they were employed there. This frequently led to marital instability as well as the loss of support from spouses and other key individuals [46, 47]. Furthermore, with this increased prevalence of marital dissolution, women entered the workforce in larger numbers to support their families, often shouldering greater financial and familial burdens due to diminished support from their husbands [46, 48]. For example, the general absence of middle-aged men from households due to labour migration often left older women with more responsibilities for supporting their extended family [38, 49, 50]. In turn, the additional demands of becoming the primary caregiver can lead to heightened psychological distress, which can be further compounded by economic hardship or poverty [51]. Importantly, previous research has found differences by gender in the relationship between contact frequency with close family, friends, neighbors, and health, meaning that men and women may experience the downstream effects of social support differently [52]. Therefore, we hypothesize that greater social support receipt will be associated with increased mortality among women compared to men across all four domains given the increase in competing demands as a result of shifting norms around living arrangements and care responsibilities caused by marital dissolution and labour migration.

Third, the country as a whole is rapidly aging with the number of individuals 65 years of age or older projected to triple by 2060 [53]. This demographic and epidemiologic transition, as well as the overall economic situation in the country, have also impacted social support for

aging individuals as well. Past research has found that age can have a moderating effect between social support and longevity meaning that social support likely impacts health and mortality in different ways for younger individuals compared to older individuals [54]. For example, in South Africa, many older adults are left to head the household while their younger family members travel for work as a result of high unemployment rates. While this is partially due to the unique labour migration patterns in rural areas, it is also because all adults over 60 years of age become eligible for a small government pension, meaning those who receive it often become the primary source of income for the household [38, 55]. In turn, such grants can create expectations that individuals will be compensated with financial resources in return for supporting older family members with pensions [56–58]. Although this has shifted traditions and norms around who cares for family members, it has also changed social support patterns for aging individuals by creating heightened demands for support through the pension funds, which are often the only steady cash income source for the household. [38, 49, 50, 59, 60]. Overall, post-Apartheid labour and economic policies have altered the social landscape of caregiving for older individuals in many rural households which, by extension, also shifted additional support demands onto aging family members as well [61, 62]. Therefore, we hypothesize that greater receipt of social support will be associated with higher mortality among older individuals compared to younger respondents across all four domains due to heightened demands, distress, and care obligations to look after the rest of the household using funds from the pension scheme.

Together, these wide disparities between the rich and poor, circular labour migration patterns between rural and urban areas, and changing ideals around social support have generated distinctive social support systems which influence the informational, emotional, and material support that rural residents receive [40, 63]. This is important since informal aid, especially via social ties, has been shown to buffer against poverty-associated stress in the United States [64, 65]. However, such associations would likely carry across into other economic settings as well. Yet, there is very little evidence regarding the relationship between social support and mortality in lower- and middle-income settings in general, and sub-Saharan Africa in particular. It is therefore vital to understand the relationship between informal aid and living well, especially in the Agincourt area where formal public-sector aid is limited.

## 2. Materials & methods

### 2.1 Ethics statement

This study received ethical approval from the Institutional Review Board at the Harvard School of Public Health (Protocol: IRB13-1608), as well from the Human Research Ethics Committee at the University of Witwatersrand (Protocol: M141159). Informed consent was collected via verbal and written consent forms administered to all respondents prior to participation in the study.

### 2.2 Study site & cohort background

Health and Aging in Africa: a Longitudinal Study of an INDEPTH community in South Africa (HAALSI), is a population-based cohort study of health and aging which collects demographic and economic information among an older population living in rural Mpumalanga Province, South Africa [66]. HAALSI is the first Health and Retirement Study sister study conducted in sub-Saharan Africa. It is nested within the Agincourt Health and Socio-Demographic Surveillance Site (HDSS), which has been run by the MRC/Wits Rural Public Health and Health Transitions Research Unit (Agincourt) since 1992, and conducts an annual census capturing data on vital events on a population of about 117,000 people [66]. Although life expectancies

have improved in Agincourt, the legacy of Apartheid still affects the community via underdeveloped and underfunded public works like education, electricity, and water [67]. Despite this, demographics of the HAALSI cohort remain similar to those in other parts of rural South Africa [68, 69].

The HAALSI cohort sampled individuals aged 40 years of age or older in July 2014 from the Agincourt HDSS [67] and 5,059 individuals completed baseline wave interviews between November 1st, 2014 and November 30th, 2015. A second wave of data collection was conducted between November 1st, 2018 and November 30th, 2019 and was completed by 4,176 (82.6%) respondents. Between waves, 595 respondents (11.8%) died, 254 individuals (5%) refused to participate in the wave two interview, 31 respondents (0.6%) were not locatable, and three respondents (0.06%) had incomplete interviews. Survey data were collected in the local language, Shangaan, via computer assisted personal interviewing (CAPI).

### 2.3 Measures

**2.3.1 Outcome.** Our primary outcome was all-cause mortality. Baseline respondents were tracked between waves through follow-up calls conducted every six months. In the event a respondent did not answer their phone, additional calls with family members, friends, or close neighbors were used as a back-up to track participants and their vital status. Date and cause of death was either obtained during these calls, determined separately by a verbal autopsy team, reported by a family member during wave two data collection, discovered during visits for other Agincourt studies, or taken from the HDSS. Loss-to-follow-up date was defined as either the date on which a given individual withdrew from the study (refusals), or the last date of contact for those who could not be found at wave two. Finally, any individuals who were alive five years after enrollment were right-censored in the analysis.

**2.3.2 Exposures.** Social support was measured by capturing respondents' "egocentric social networks" [38, 70, 71]. In this process, respondent's ("egos") were asked to name the six most important individuals ("alters") with whom they have been in contact with over the past six months; spouses were automatically added as a seventh alter if not otherwise named. Respondents were asked how often each alter provided them with informational, emotional, physical, and financial support over the past six months on a seven-category scale. Emotional support was assessed as the frequency respondents typically received support from each alter, "...such as when you are feeling sad or anxious or upset"; 2. Physical support—"...such as when you have needed help with chores around the house or at work, taking care of yourself or going from one place to another"; 3. Informational support—"...such as receiving advice about important health issues, employment issues, or any other important matters"; and, 4. Financial support—"...such as borrowing money, receiving food, being given a job or anything else related to money or in-kind transfers". We subsequently recoded answers to approximate monthly contact frequencies (values in parentheses): almost every day (30); a few times per week (10); once per week (4); a few times per month (2); once per month (1); less than once a month (0). Additionally, fourteen respondents either refused to list, or did not know any alters, and were therefore coded as contributing zero person-days. These values were then summed across all alters to yield a score indicating each respondents' total number of person-days of social support received in a month for a given support type. Finally, to increase interpretability, we standardized each social support measure by dividing the number of person-days by its standard deviation thereby creating Z-scores within the study sample [38].

**2.3.3 Covariates.** We considered a set of baseline covariates likely to predict both social support and mortality in this population. Furthermore, these measures were included in this analysis as potential confounders of the social support and mortality relationship, rather than

as targets for inference in and of themselves. Socio-demographic variables included categorical age (in decades, to allow for non-linearities in effect), sex, marital status (never married; currently cohabiting/married; separated/divorced; widowed), employment status (full/part time employed; homemaker; not employed), pension receipt, natal country (South Africa; other), household wealth (in quintiles), respondent educational attainment (none; any primary; any secondary; completed secondary or more), literacy status (yes; no), and paternal education (none; any; unknown). Self-reported health measures included depression (CESD-8 continuous measure) [72], post-traumatic stress disorder (PTSD; > 4 of 7 symptoms on DSM-IV screening scale) [73], cognitive status (0–26 continuous scale on four separate memory tests: immediate word recall (10 points); delayed word recall (10 points); orientation (4 points); and numeracy (2 points)) [74], limitations in Activities of Daily Living (ADLs; any vs. none) [75], and frailty index (non-frail; pre-frail; frail; unable to score) [76, 77].

Several covariates were derived from biomarkers or field-based measurements, since health conditions may affect social support levels—either rising due to greater support needs or falling due to diminished capacity to interact. Missing values were, where possible, coded into those actively declining to participate in measurement, and those with data missing due to processing errors, since the missingness mechanism is likely to differ for these groups. HIV status was defined as positive or non-positive (including indeterminate) from dried blood spot (DBS) assay tests as previously described [66]. Anemia was categorized following the South African National Health and Nutrition Examination Survey (SANHANES) guidelines–(Men: Normal >12.9g/dl; Mild anemia ≤12.9g/dl and ≥11g/dl; Moderate anemia <11g/dl and ≥8g/dl; Severe anemia <8g/dl; Women: Normal >11.9g/dl; Mild anemia ≤11.9g/dl and ≥11g/dl; Moderate anemia <11g/dl and ≥8g/dl; and Severe anemia <8g/dl) [78]. Respondents were considered as having diabetes if they self-reported ever being diagnosed, had fasting glucose levels ≥7 mmol/l (126 mg/dL), or non-fasting glucose levels ≥11.1 mmol/l (200 mg/dL) [79]. Respondents were considered hypertensive if they had: i) systolic blood pressure ≥140 mmHg; ii) diastolic blood pressure ≥ 90 mmHg, or iii) reported using anti-hypertensive medication at baseline interview [78, 80]. Body mass index (BMI) was defined according to the World Health Organization's adult BMI classification guidelines based on field-based measurements [81].

**2.3.4 Statistical analysis.** We included all respondents who had complete data on the exposure and outcome, as well as covariates. Descriptive statistics were generated for all variables of interest, and we reported means and standard deviations for continuous variables, as well as proportions for categorical variables. Differences by sex were assessed using two-sample t-tests for continuous variables and Pearsons' chi-squared tests for categorical variables. We used the Kaplan-Meier method to estimate the outcome, with appropriate right-censoring to estimate survival time from baseline interview to follow-up interview date, death, lost-to-follow-up, or refusal to complete a follow-up interview. Separate Cox proportional hazard survival regression models were run in the presence of each of the four social support domains. Assumptions of proportionality were assessed and verified using both Kaplan Meier curves and scaled Schoenfeld residual plots, with each support domain first transformed into quintiles for ease of interpretation.

All models were adjusted for categorical age, sex, marital status, employment status, educational status, pension receipt, and biomarker measures, using an indicator for missing covariate values given the limited level of missingness [24, 79]. Domain-specific models were further adjusted as follows: (1) informational support—respondent natal country, and literacy; (2) emotional support—depression and PTSD; (3) financial support—household wealth, and paternal education; (4) physical support—cognitive score, ADL limitations, and frailty. We also considered whether the associations differed by sex, and by age (under/over age 60) by

adding interactions of the exposure and these variables in separate models, testing the interaction terms with Wald tests. Hazard ratios for the main effects and for interaction effects are reported for individual models. We ran three sensitivity checks on our results, first excluding all health event measures, second using parametric accelerated failure time models with exponential distributions, and third using dichotomous transformations of each support domain.

## 3. Results

After removing 116 cases due to proxy-respondent interviews (who were not asked social network questions), and 36 other cases due to missing outcome values (interview date errors), our analytic sample included 4,907 individuals (Table 1). Missingness was under 10% for all covariates. Respondents' mean age was 62.2 years and 53.7% of the sample were female. Most participants were either currently married (51.8%) or widowed (29.9%). Just over two-thirds of the men were married at the time of interview (69.3%), compared to just over one-third of women (36.8%). Around 70% of both men and women in the sample were born in South Africa (71.5% and 69.4%, respectively). Most respondents (73.2%) were unemployed with marginal differences between males and females (72.5% vs. 73.8%). Almost half of the sample (44.9%) had no formal education. One-fifth (21%) of respondents were living with HIV, and 11% of respondents were considered as having diabetes. Most respondents were either normal weight (34.1%) or obese (27.9%). More males had a normal body mass index (BMI) compared to females (43.6% vs. 25.9%), while the reverse was true for obese participants (14.9% vs. 39.2%). A majority of respondents (62.3%) were considered hypertensive, with 57.2% of males reporting high blood pressure, compared to two-third of females (66.7%). Mean cognition scores were 12.4, with males reporting slightly higher scores (13.0), compared to females (11.9).

Respondents received approximately one support event per day for informational (mean monthly events = 30.1, standard deviation (SD) = 33.3), emotional (mean = 27.2, SD = 33) and physical (mean = 25.1, SD = 26.1) support. Financial support was less common, with a mean of 15.4 person-days per month (SD = 22.4). Across all four support domains, males had higher average support events per day compared to females, with physical and financial support showing larger differences between the sexes. Between study waves 517 respondents in our sample (10.6% of 4,907) died. Of those who died, 57.6% were male. Fig 1 shows that five years after completing a baseline interview, females had approximately 87% chance of survival and males had approximately 84%.

In our adjusted regression models, each social support variable showed small positive associations with mortality (Table 2). Hazard ratios per standard deviation of outcome ranged from 1.04 (95% confidence interval [CI]: 0.95,1.13) for financial support, to 1.09 (95% CI: 0.99,1.18) for informational support. Effect modification with age and sex mostly showed stronger associations for females compared to men, and individuals under 60 years of age compared to those greater than or equal to 60 years of age, respectively. Hazard ratios for females ranged from 1.06 (95% CI: 0.93,1.21) for physical support to 1.13 (95% CI: 1.00,1.29) for informational support, while hazard ratios for individuals under 60 years of age ranged from 0.95 (95% CI: 0.78,1.16) for financial support, to 1.15 (95% CI: 0.97,1.35) for informational support.

This table reports key coefficients from twelve regression models: three for each support domain. All values are Hazard Ratios and [95% confidence intervals] from models adjusted for all covariates listed in S1 Table. Wald tests were used to assess significance of interactions. Full models are provided in S1–S3 Tables. N for all models is 4907. The first row of sections B & C is an indicator variable of Males (reference) vs. Females, or ≥ 60 (reference) vs. < 60. The second and third rows of sections B & C show the effect of a single unit increase of a given support

**Table 1. Descriptive statistics of analytic sample, stratified by sex.**

|  | Overall | Female | Male | Test Statistic* | p-value |
|---|---|---|---|---|---|
| Male | 46.3% |  |  |  |  |
| Age |  |  |  | 239 | <0.001 |
| 40–49 | 17.5% | 18.1% | 16.8% |  |  |
| 50–59 | 27.6% | 28.6% | 26.4% |  |  |
| 60–69 | 26.2% | 24.8% | 27.9% |  |  |
| 70–79 | 17.6% | 16.2% | 19.3% |  |  |
| 80+ | 11.0% | 12.3% | 9.6% |  |  |
| Marital status |  |  |  | 22.4 | <0.001 |
| Never Married | 5.4% | 4.5% | 6.5% |  |  |
| Currently married | 51.8% | 36.8% | 69.3% |  |  |
| Separated/Divorced | 12.8% | 13.0% | 12.6% |  |  |
| Widowed | 29.9% | 45.7% | 11.6% |  |  |
| Wealth quintile |  |  |  | 6.0 | 0.20 |
| Lowest | 20.3% | 19.9% | 20.7% |  |  |
| Second highest | 19.7% | 20.1% | 19.3% |  |  |
| Middle | 19.7% | 19.9% | 19.4% |  |  |
| Second highest | 20.0% | 20.4% | 19.6% |  |  |
| Highest | 20.3% | 19.7% | 21.0% |  |  |
| Born in South Africa | 70.3% | 69.4% | 71.5% | 0.02 | 0.89 |
| Educational attainment |  |  |  | 59.1 | <0.001 |
| No formal | 44.9% | 48.9% | 40.2% |  |  |
| Some primary | 34.4% | 33.3% | 35.8% |  |  |
| Some secondary | 11.6% | 9.8% | 13.7% |  |  |
| Completed secondary | 9.1% | 8.1% | 10.3% |  |  |
| Respondent Employment |  |  |  | 40.4 | <0.001 |
| Employed | 16.3% | 13.7% | 19.4% |  |  |
| Unemployed | 73.2% | 73.8% | 72.5% |  |  |
| Homemaker | 10.5% | 12.6% | 8.1% |  |  |
| Any literacy | 59.5% | 51.8% | 68.6% | 53 | <0.001 |
| Father's education |  |  |  | 6.4 | 0.04 |
| None | 78.7% | 77.4% | 80.4% |  |  |
| Any | 13.5% | 14.1% | 12.7% |  |  |
| Missing data | 7.8% | 8.5% | 7.0% |  |  |
| Receiving state pension | 34.6% | 32.2% | 37.4% | 58.3 | <0.001 |
| HIV serostatus |  |  |  | 1.8 | 0.41 |
| Positive | 21.0% | 21.2% | 20.8% |  |  |
| Negative | 70.1% | 70.7% | 69.3% |  |  |
| Missing data | 8.9% | 8.2% | 9.9% |  |  |
| Anemia status |  |  |  | 52.6 | <0.001 |
| None | 52.3% | 51.3% | 53.5% |  |  |
| Mild | 21.6% | 18.4% | 25.5% |  |  |
| Moderate | 13.4% | 17.6% | 8.5% |  |  |
| Severe | 1.9% | 2.3% | 1.3% |  |  |
| Declined test | 6.7% | 6.2% | 7.2% |  |  |
| Processing error | 4.1% | 4.2% | 4.1% |  |  |
| Blood pressure |  |  |  | 15.8 | <0.001 |
| Hypertensive | 62.3% | 66.7% | 57.2% |  |  |
| Normotensive | 35.8% | 32.0% | 40.3% |  |  |

*(Continued)*

**Table 1.** (Continued)

| | Overall | Female | Male | Test Statistic* | p-value |
|---|---|---|---|---|---|
| Declined test | 1.6% | 1.0% | 2.2% | | |
| Processing error | 0.3% | 0.3% | 0.3% | | |
| Body mass index | | | | 193 | <0.001 |
| Underweight | 5.0% | 2.5% | 7.9% | | |
| Normal | 34.1% | 25.9% | 43.6% | | |
| Overweight | 26.6% | 26.6% | 26.6% | | |
| Obese | 27.9% | 39.2% | 14.9% | | |
| Processing error | 6.4% | 5.8% | 7.0% | | |
| Diabetic | 11.0% | 12.0% | 9.9% | 18.5 | <0.001 |
| Missing on Diabetes | 7.7% | 7.2% | 8.2% | 0.01 | 0.91 |
| Cognitive score* | 12.4 (5.3) | 11.9 (5.4) | 13.0 (5.1) | 10.3 | <0.001 |
| Cognitive Test not completed | 0.5% | 0.4% | 0.6% | 0.1 | 0.75 |
| Frailty status | | | | 202 | <0.001 |
| Non-frail | 51.5% | 51.5% | 51.6% | | |
| Pre-frail | 36.8% | 37.7% | 35.7% | | |
| Frail | 3.0% | 2.7% | 3.4% | | |
| Unable to score | 8.7% | 8.1% | 9.3% | | |
| Depression screen positive* | 1.4 (1.6) | 1.5 (1.7) | 1.4 (1.5) | -7.5 | <0.001 |
| PTSD positive | 2.8% | 3.2% | 2.3% | 0.9 | 0.33 |
| Any Activities of Daily Living | 8.0% | 8.0% | 8.1% | 119 | <0.001 |
| Informational support* | 30.1 (33.3) | 29.1 (33.6) | 31.3 (32.9) | -0.9 | 0.39 |
| Emotional support* | 27.2 (33.0) | 26.3 (32.9) | 28.2 (33.1) | -1.7 | 0.10 |
| Physical support* | 25.1 (26.1) | 21.9 (25.1) | 28.8 (26.8) | -1.7 | 0.10 |
| Financial support* | 15.4 (22.4) | 13.5 (21.5) | 17.5 (23.2) | -1.2 | 0.24 |

Tests are Pearson's chi-squared of percentages for categorical variables unless otherwise stated;

*continuous variables use a two-sample t-test of means; continuous variables presented as: mean (SD)

type for one group, and a single unit increase in a given support type for each gender and age-group, respectively.

We ran several sensitivity analyses. First, given the possibility that some of the variables could act as mediators, we tested models that excluded all health event measures (PTSD, cognition, depression, ADL limitations, and frailty status) to assess whether they affected the results. Results were statistically similar regardless of whether these variables were included or not (S4–S7 Tables). Second, results from the accelerated failure time models were also nearly identical to the Cox proportional hazard models, suggesting that our findings do not rely on proportionality assumptions (S8–S11 Tables).

A final sensitivity analysis was run with dichotomous social support variables replacing the continuous ones (S12–S15 Tables). These were classified by whether any of a given respondent's alters provided support at least once per month. Main effects from the non-interaction models yielded larger magnitudes ranging from 0.86 [0.69,1.08] for financial support, to 1.31 [0.93,1.85] for informational support. Notably, the coefficient for financial support switched directions and became protective. Effect modification with sex and age demonstrated similar positive associations, though again with larger magnitudes, for females and individuals under 60 years of age compared to males and those greater than or equal to 60 years of age, respectively.

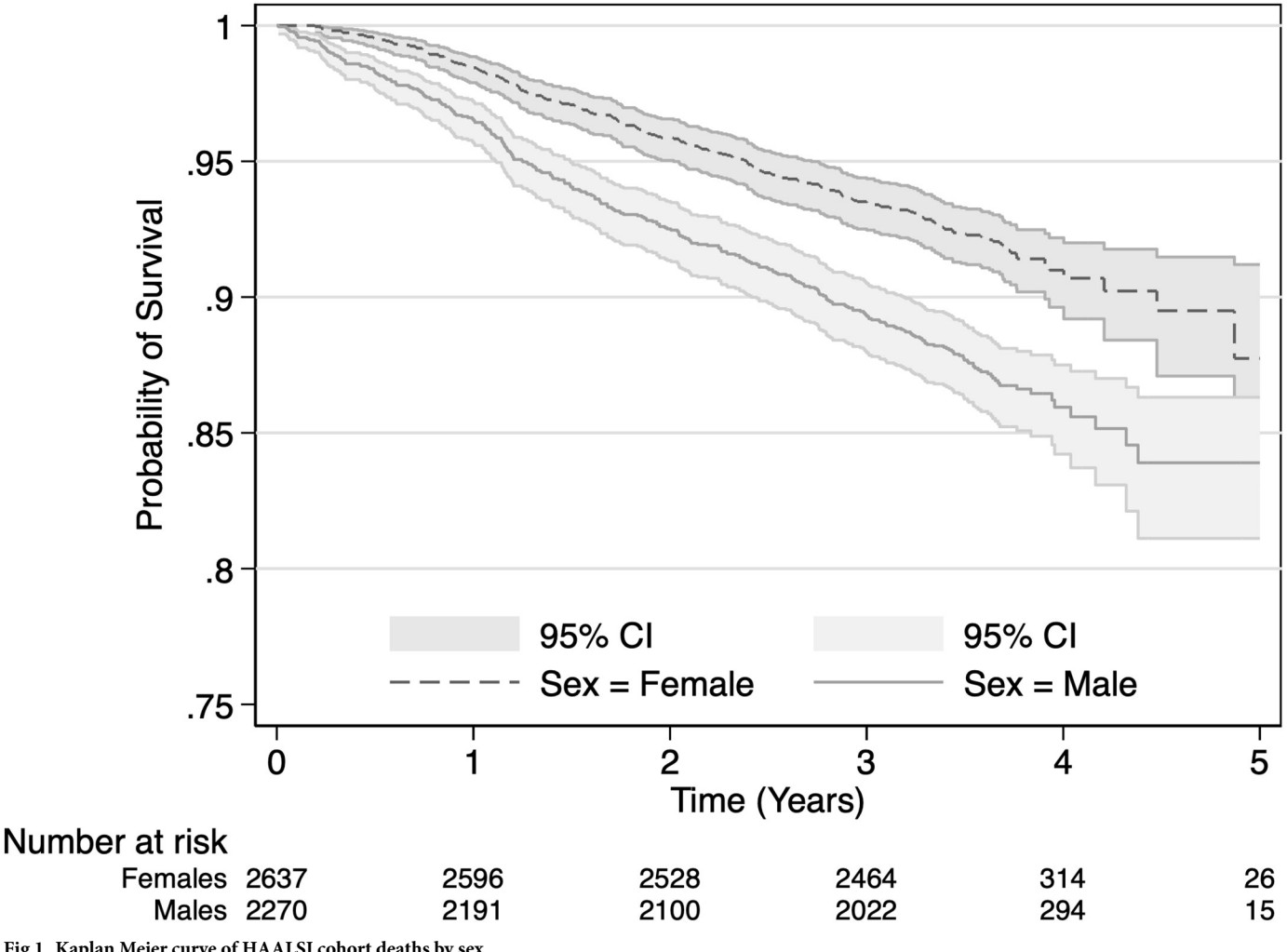

**Fig 1. Kaplan Meier curve of HAALSI cohort deaths by sex.**

**Table 2. Adjusted cox proportional hazard models for mortality in HAALSI between waves one and two, by presence of social support domains.**

| Support type | Informational | | Emotional | | Financial | | Physical | |
|---|---|---|---|---|---|---|---|---|
| A. Main effects only | | | | | | | | |
| Social support | 1.09 | [0.99,1.18] | 1.08 | [1.00,1.18] | 1.04 | [0.95,1.13] | 1.07 | [0.98,1.16] |
| B. Sex and support interaction | | | | | | | | |
| Males vs. females | 2.14 | [1.72,2.66] | 2.05 | [1.65,2.54] | 2.05 | [1.65,2.54] | 2.06 | [1.65,2.56] |
| Social support in females | 1.13 | [1.00,1.29] | 1.10 | [0.97,1.25] | 1.10 | [0.97,1.25] | 1.06 | [0.93,1.21] |
| Social support in males | 1.05 | [0.94,1.18] | 1.07 | [0.96,1.19] | 1.00 | [0.89,1.12] | 1.07 | [0.96,1.20] |
| $\chi^2$ for interaction | 0.75 | | 0.12 | | 1.45 | | 0.03 | |
| p-value | 0.39 | | 0.73 | | 0.23 | | 0.86 | |
| C. Age and support interaction | | | | | | | | |
| $\geq$ 60 vs. < 60 | 6.70 | [4.05,11.07] | 6.80 | [4.12,11.22] | 7.01 | [4.24,11.58] | 5.83 | [3.51,9.67] |
| Social support in those < 60 | 1.15 | [0.97,1.35] | 1.11 | [0.93,1.31] | 0.95 | [0.78,1.16] | 1.10 | [0.93,1.31] |
| Social support in those $\geq$ 60 | 1.06 | [0.96,1.18] | 1.08 | [0.98,1.18] | 1.06 | [0.97,1.17] | 1.06 | [0.96,1.16] |
| $\chi^2$ for interaction | 0.58 | | 0.07 | | 1.07 | | 0.17 | |
| p-value | 0.45 | | 0.79 | | 0.30 | | 0.68 | |

## 4. Discussion

In a cohort of rural South Africans aged 40 and above, we found positive associations between baseline social support and subsequent mortality over an average of four years. Receipt of each social support type was modestly positively associated with mortality. The magnitude of association of support per standard deviation was similar to the effect of receiving a pension, being hypertensive, or screening positive for depression (in the case of emotional support domain). This core finding of a positive association with mortality contrasts with results from other settings, including elsewhere in Africa [82–86]. For example, a worldwide meta-analytic review of 148 studies from Europe, North America, and Asia found that receipt of emotional, informational, tangible, or belonging support was non-significantly positively associated with survival [11]. Furthermore, our hypothesis that financial support would be the most predictive of mortality across each of the four domains due to the resource-constrained nature of Agincourt was not supported by the results either. In fact, financial support had the least positive hazard of mortality, though with the caveat that this particular result was statistically non-significant. Additionally, our finding that associations per standard deviation of support were somewhat less positive for men compared to women also contrasts with most, though not all, prior research [17, 87, 88]. However, our results generally do demonstrate that across each of the four domains of social support, older individuals have a higher association with mortality than younger ones. There are several possible explanations for our primary results, which we consider in turn. Importantly, some of these explanations call for supplementing measures of interaction frequency with those of interaction quality to generate a fuller picture how social support may affect mortality in this setting.

First, we may be oversimplifying the relationship between social support and mortality. For example, the effect of social support may be heterogeneous, such that some individuals benefit while others are harmed. While certain respondents within HAALSI receive a great deal of support relative to the rest of the cohort, surplus support has sometimes been shown to be detrimental to health [89, 90]. An excess of support to older individuals from their younger family members might increase distress by limiting independence, particularly if it happens following unexpected illness [91, 92]. Furthermore, this can subsequently lead to increased dependence and further deterioration as well [91, 92]. Alternatively, higher quantities of support could cause distress due to reciprocal expectations of social support provision, especially in settings where social support is used as a social safety net in lieu of material and financial resources [37]. For example, in some low-income US communities, individuals can inadvertently cause one-another psychological distress through excessive social demands [51]. This can occur in high-income settings as well where women in particular are expected to give emotional support [93, 94]. These implicit expectations of support reciprocity could have specific negative implications for elderly HAALSI participants. For instance, the prominence of non-contributory state payments to older adults—notably the pension grant—in poor rural South African economies can lead to the expectation that individuals who support elderly family members will be rewarded with financial support in return [56–58]. Therefore, individuals receiving more care may face competing demands for such financial support and may, as a result, have less leftover for themselves. These mechanisms may have countervailing effects, cancelling one-another out in this population as well. Given that social support can have varying dimensions and directions, further research would be useful to unpick how support levels are perceived within the community.

Second, another form of heterogeneity driving the association between social support and mortality might be present: while social support can protect against physical and cognitive decline, it can also increase (due to greater need) or decrease (due to increased difficulty

maintaining social relationships) as a result of prior ill-health [71, 95]. This is especially true of the elderly or other frail individuals whose needs increase over time [96]. In HAALSI, although the data is longitudinal, some participants with high baseline levels of social support may have had these levels because they were already ill, frail, or otherwise in need of care [97–99]. While our approach builds on other longitudinal research which has shown how social support can affect mortality, especially amongst lower socioeconomic status groups, we cannot rule out the effects of pre-baseline morbidity on social support (among at least some participants) which would confound our analysis. Given the longitudinal nature of this study, revisiting these analyses using multiples waves of social support data once available could help to tease out the causal nature of the support-mortality pathway.

Third, the quantity of social support received might not be high enough for anybody in this setting. Despite the appearance of a weakly positive association amongst support levels within HAALSI, it is possible that very few respondents, if any, have reached some latent threshold of support necessary to protect against mortality. Therefore, even participants with relatively high levels of support in the cohort might still not be receiving enough for the effects to register as protective. Consequently, this could make it seem like social support is harmful, since even respondents with the highest quantities have increased mortality. Alternatively, another possible explanation for support levels being insufficient is that some other factor, like economic wellbeing, could be more predictive of mortality in this setting. For example, income and wealth typically affect health strongly up to a certain level, above which additional resources are increasingly marginal [100–102]. Therefore, it is possible social support could impact health and mortality in a similar format in this context once basic needs are met, but the poverty of the area has prevented anyone from reaching said threshold. Indeed, it is possible that individuals under this threshold, but relatively well off in the community, could have worse outcomes due to increased demands for financial and other support as outlined above. Therefore, determining whether subjective support levels are perceived to be sufficient, or capturing objective measures of support quality, would help in assessing such an 'insufficiency' hypothesis.

Finally, social support might truly not matter for mortality in this setting. This could be because other factors (e.g., insufficient material resources, government non-contributory grants, poor mental health) are more important for disease acquisition, ill-health and death regardless of social support level. Alternatively, given that the most common causes of death in Agincourt—notably dementia, cancer, and HIV—differ substantially from those in higher-income settings where social support and mortality have previously been analysed, social support may not play an important role for these conditions [69, 103]. Such a hypothesis could be tested by comparing HAALSI data with that of other communities of varying income levels through comparable surveys.

## 4.1 Limitations

Given observational data, our findings are always open to unmeasured confounding. While our analytic design was informed by prior research, the specifics of this setting may mean we missed potential confounders. Secondly, although the HAALSI survey captures frequency of support receipt, it could be that another measurement of support is predictive of mortality in this setting. For example, some research has found the provision of social support to be more beneficial to longevity than receipt [104], while other research suggests that a healthy balance of both giving and receiving was associated with lower mortality than higher levels of either one alone [105]. Additionally, the relationship of the individual providing support could matter as well, with younger individuals only receiving mental health benefits if the aid came from

a partner [106]. Alternatively, support quality may be the key determinant. HAALSI's focus on social support quantities is in-line with the typical simplifying assumption that social relationships are homogeneously positive [11]. However, negative interactions have been theorized to be more predictive of mortality risk than low quality, positive ones [19–21]. Our use of counts of social support events may hide a mixture of receipt, method, and quality of social support—attenuating any true association in our analysis. If these are also the case in South Africa, then relying on measures of receipt alone—as in HAALSI—could miss key effects. Furthermore, our social support measures were self-reported as well, and therefore may be subject to recall and social desirability bias, the direction of which is hard to assess. Finally, given the necessarily relational nature of our research question, it is difficult to be sure how generalizable our findings are—especially since they appear to differ from other South African data. Nevertheless, it is likely to have applicability to other settings with limited resources, particularly in relation to formal care and support services. Additional data from other rural and urban low-income settings will allow triangulation of results and thus clearer understanding of the generalizability of this work.

## 5. Conclusion

Using data from a large cohort of adults aged 40 and above living in rural South Africa, we found generally positive albeit not statistically significant associations between levels of social support and mortality. These results suggest social support may not be a major factor affecting mortality in this cohort—although support is likely important to wellbeing, other factors may be more important drivers of death in this setting. Alternatively, it may be that a combination of measurement approaches are needed to determine how quantity, quality, mode, and continuity of support jointly affect the health of older adults in rural South Africa.

## Supporting information

**S1 Table. Cox proportional hazard models, full—No interaction.**
(PDF)

**S2 Table. Cox proportional hazard models, full—Sex interaction.**
(PDF)

**S3 Table. Cox proportional hazard models, full—Age interaction.**
(PDF)

**S4 Table. Adjusted Cox proportional hazard models for mortality in HAALSI between waves one and two, by presence of social support domains—(without health events).**
(PDF)

**S5 Table. Cox proportional hazard models, full—No interaction—(without health events).**
(PDF)

**S6 Table. Cox proportional hazard models, full—Sex interaction—(without health events).**
(PDF)

**S7 Table. Cox proportional hazard models, full—Age interaction—(without health events).**
(PDF)

**S8 Table. Adjusted accelerated failure time hazard models for mortality in HAALSI between waves one and two, by presence of social support domains.**
(PDF)

**S9 Table. Accelerated failure time hazard models, no interaction.**
(PDF)

**S10 Table. Accelerated failure time hazard models, sex interaction.**
(PDF)

**S11 Table. Accelerated failure time hazard models, age interaction.**
(PDF)

**S12 Table. Adjusted Cox proportional hazard models for mortality in HAALSI between waves one and two, by presence of social support domains—(Dichotomous support).**
(PDF)

**S13 Table. Cox proportional hazard models, full—No interaction—(Dichotomous support).**
(PDF)

**S14 Table. Cox proportional hazard models, full—Sex interaction—(Dichotomous support).**
(PDF)

**S15 Table. Cox proportional hazard models, full—Age interaction—(Dichotomous support).**
(PDF)

**S1 Fig. Kaplan Meier curve of informational support quintiles.**
(TIF)

**S2 Fig. Kaplan Meier curve of emotional support quintiles.**
(TIF)

**S3 Fig. Kaplan Meier curve of physical support quintiles.**
(TIF)

**S4 Fig. Kaplan Meier curve of financial support quintiles.**
(TIF)

**S5 Fig. Scaled Schoenfeld Residuals—Informational support (Quintiles).**
(TIF)

**S6 Fig. Scaled Schoenfeld Residuals—Emotional support (Quintiles).**
(TIF)

**S7 Fig. Scaled Schoenfeld Residuals—Physical support (Quintiles).**
(TIF)

**S8 Fig. Scaled Schoenfeld Residuals—Financial support (Quintiles).**
(TIF)

**S1 Checklist. Inclusivity in global research.**
(DOCX)

## Acknowledgments

The authors acknowledge the invaluable support provided by the MRC/Wits Rural Public Health and Health Transitions Research Unit for their work overseeing data collection, management, operations, processing, and administration in Agincourt, South Africa.

## Author Contributions

**Conceptualization:** David Kapaon, Guy Harling.

**Data curation:** David Kapaon, Carlos Riumallo-Herl, Chodziwadziwa Whiteson Kabudula.

**Formal analysis:** David Kapaon, Guy Harling.

**Funding acquisition:** Carlos Riumallo-Herl, Guy Harling.

**Investigation:** David Kapaon, Guy Harling.

**Methodology:** David Kapaon, Carlos Riumallo-Herl, Elyse Jennings, Shafika Abrahams-Gessel, Keletso Makofane, Chodziwadziwa Whiteson Kabudula, Guy Harling.

**Project administration:** Elyse Jennings, Shafika Abrahams-Gessel, Guy Harling.

**Resources:** Guy Harling.

**Supervision:** Carlos Riumallo-Herl, Elyse Jennings, Guy Harling.

**Visualization:** David Kapaon.

**Writing – original draft:** David Kapaon, Elyse Jennings, Guy Harling.

**Writing – review & editing:** David Kapaon, Carlos Riumallo-Herl, Elyse Jennings, Shafika Abrahams-Gessel, Keletso Makofane, Chodziwadziwa Whiteson Kabudula, Guy Harling.

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
