## [Decision Letter · Decision Letter 0]

25 Mar 2024

PGPH-D-23-02132

Social support receipt as a predictor of mortality: a cohort study in rural South Africa

Dear Dr. Kapaon,

Thank you for submitting your manuscript to PLOS Global Public Health. After careful consideration, we feel that it has merit but does not fully meet PLOS Global Public Health’s publication criteria as it currently stands. Therefore, we invite you to submit a revised version of the manuscript that addresses the points raised during the review process.

We look forward to receiving your revised manuscript.

Kind regards,

Daniel Kim, M.D., Dr.P.H.

Academic Editor

Journal Requirements:

3. Please send a completed 'Competing Interests' statement, including any COIs declared by your co-authors. If you have no competing interests to declare, please state "The authors have declared that no competing interests exist". Otherwise please declare all competing interests beginning with the statement "I have read the journal's policy and the authors of this manuscript have the following competing interests:"

4. Please provide separate figure files in .tif or .eps format only and remove any figures embedded in your manuscript file. Please also ensure all files are under our size limit of 10MB.

Additional Editor Comments (if provided):

Reviewers' comments:

Reviewer's Responses to Questions

**Comments to the Author**

1. Does this manuscript meet PLOS Global Public Health’s publication criteria? Is the manuscript technically sound, and do the data support the conclusions? The manuscript must describe methodologically and ethically rigorous research with conclusions that are appropriately drawn based on the data presented.

Reviewer #1: Yes

Reviewer #2: Yes

2. Has the statistical analysis been performed appropriately and rigorously?

Reviewer #1: Yes

Reviewer #2: Yes

3. Have the authors made all data underlying the findings in their manuscript fully available (please refer to the Data Availability Statement at the start of the manuscript PDF file)?

Reviewer #1: Yes

Reviewer #2: No

4. Is the manuscript presented in an intelligible fashion and written in standard English?

Reviewer #1: Yes

Reviewer #2: Yes

5. Review Comments to the Author

Reviewer #1: Dear Authors,

Thanks for submitting this manuscript. You tackle an important topic (social support among older people), especially in the reported context (low-income, rural, South Africa). Given the paucity of research, this article will further the knowledge and evidence base in the topic, and the recommended further research, given the unique findings is welcome. The manuscript, though discussing a complex topic with equally complex statistical tests, is also presented in an easy to understand manner, which makes it more likely to be accessed by interested stakeholders. I enjoyed reading the manuscript. Well done.

Reviewer #2: Social support receipt as a predictor of mortality: a cohort study in rural South Africa

I appreciated reading your important work on social support in rural South Africa. It is very timely as demographics in Africa are shifting with increased life expectancy and the rural-urban migration in search for job opportunities.

Specific comments

1. Since almost half of the sample (44.9%) had no formal education and 34.4% had some primary education. What language was used for the survey in this rural community? Could this have had an impact on responses to some of the survey questions or their understanding of what was asked? I reviewed the Cohort profile (Gómez-Olivé et al., 2018) and it only mentions that xiTsonga or English was used for obtaining informed consent but not the survey itself.

2. Center for Epidemiological Studies-Depression (CES-D) Instrument.

I appreciate the measure of depression in this study. However, in most African cultures, the term "depression" is uncommon. How were questions 3 and 6 translated? Was there a pilot study for this CES-D instrument to check for its application in this setting?

3. African cultures have traditional practices where it is expected that the elders are respected and are taken care of by the younger generations even though this practice is rapidly changing with the movement to the urban areas in search of employment. Do you think this has any implications for your study?

4. It is commendable that this study measured blood pressure and took participant's height and weight measurements for computing BMI. The overall prevalence for those who were hypertensive was 62.3%, with 66.7% of women being hypertensive versus 57% of males. In addition, 54.5% were overweight or obese. These findings, in a resource-constrained setting, are disturbing. Even though this was not the main focus of the current study, what urgent interventions would you propose to address these risk factors?

5. Since 70% were born in South Africa, were there any differences in results with those born outside of South Africa?

Other comments

Lines 48-51: A long sentence. Consider revising.

Line 155: “Third, the country…”. Where are the first and second points? Consider including these for the readers to follow the arguments presented.

Line 221: “Date and cause of death was either obtained during these calls,...” How could this have happened if someone had already passed away? Consider revising for clarity.

Line 279: “Respondents were considered diabetic” and line 324 “were considered diabetic” consider using person-first language.

Line 413. “Furthermore, this can subsequently can lead to increased” does not read well. Consider revising.

6. PLOS authors have the option to publish the peer review history of their article (what does this mean?). If published, this will include your full peer review and any attached files.

**Do you want your identity to be public for this peer review?** For information about this choice, including consent withdrawal, please see our Privacy Policy.

Reviewer #1: No

Reviewer #2: **Yes: **Adel Mburia-Mwalili

---

## [Decision Letter · Decision Letter 1]

15 Aug 2024

Social support receipt as a predictor of mortality: a cohort study in rural South Africa

PGPH-D-23-02132R1

Dear Mr. Kapaon,

We are pleased to inform you that your manuscript 'Social support receipt as a predictor of mortality: a cohort study in rural South Africa' has been provisionally accepted for publication in PLOS Global Public Health.

Best regards,

Julia Robinson

Executive Editor

Reviewer Comments (if any, and for reference):

Reviewer's Responses to Questions

**Comments to the Author**

1. If the authors have adequately addressed your comments raised in a previous round of review and you feel that this manuscript is now acceptable for publication, you may indicate that here to bypass the “Comments to the Author” section, enter your conflict of interest statement in the “Confidential to Editor” section, and submit your "Accept" recommendation.

Reviewer #2: All comments have been addressed

2. Does this manuscript meet PLOS Global Public Health’s publication criteria? Is the manuscript technically sound, and do the data support the conclusions? The manuscript must describe methodologically and ethically rigorous research with conclusions that are appropriately drawn based on the data presented.

Reviewer #2: (No Response)

3. Has the statistical analysis been performed appropriately and rigorously?

Reviewer #2: (No Response)

4. Have the authors made all data underlying the findings in their manuscript fully available (please refer to the Data Availability Statement at the start of the manuscript PDF file)?

Reviewer #2: (No Response)

5. Is the manuscript presented in an intelligible fashion and written in standard English?

Reviewer #2: (No Response)

6. Review Comments to the Author

Reviewer #2: (No Response)

7. PLOS authors have the option to publish the peer review history of their article (what does this mean?). If published, this will include your full peer review and any attached files.

**Do you want your identity to be public for this peer review?** For information about this choice, including consent withdrawal, please see our Privacy Policy.

Reviewer #2: **Yes: **Adel Mburia-Mwalili
